# “Modified Schirmer Test” as an Objective Measurement for Vaginal Dryness: A Prospective Cohort Study

**DOI:** 10.3390/diagnostics12030574

**Published:** 2022-02-23

**Authors:** Dana Gabrieli, Yael Suissa-Cohen, Sireen Jaber, Ahinoam Lev-Sagie

**Affiliations:** 1Faculty of Medicine, Hebrew University of Jerusalem, Jerusalem 9780214, Israel; levsagie@netvision.net.il; 2Department of Obstetrics and Gynecology, Hadassah-Hebrew University Medical Center, Mount Scopus, Jerusalem 9765422, Israel; yaelshi.suissa@mail.huji.ac.il (Y.S.-C.); sireenjaber@gmail.com (S.J.)

**Keywords:** genitourinary syndrome of menopause (GSM), vaginal dryness, vaginal atrophy, vaginal health index (VHI), estrogen, vaginal maturation index (VMI), fractional CO_2_ laser

## Abstract

None of the currently available parameters allow for a direct and objective measurement of vaginal moisture. We used a calibrated filter paper strip as a measurement tool for the quantification of vaginal fluid, in a similar manner as the ophthalmic “Schirmer test” (used for eye moisture measurement). The study aimed to evaluate the validity of this new, objective tool, to measure vaginal moisture. We compared vaginal moisture measurements using the “modified Schirmer test” in symptomatic women with genitourinary syndrome of menopause to those of women without vaginal dryness. The mean “modified Schirmer test” measurement in the control group was 21.7 mm compared to 3.3 mm in the study group, yielding a statistically significant difference (*p* < 0.001). Strong correlations were found between “modified Schirmer test” measurements and pH (correlation coefficient −0.714), Vaginal Health Index [VHI (0.775)], and Visual Analogue Score (VAS) of dryness during intercourse (−0.821). Our findings suggest that the “modified Schirmer test” can be used as an objective measurement for the assessment of vaginal fluid level. This test may also prove useful for evaluation of non-hormonal treatments aimed to treat vaginal dryness.

## 1. Introduction

Urogenital atrophy, also referred to as genitourinary syndrome of menopause (GSM) [1], is caused by decreased estrogen levels in women’s urogenital tissues. Symptoms include vulvovaginal discomfort described as dryness, itching, burning, irritation, and soreness; sexual dysfunction due to decreased lubrication and dyspareunia; and urinary complaints such as urgency, frequency, and recurrent urinary tract infections [2,3].

Among menopausal women, prevalence of GSM is estimated at approximately 50–60%, making it one of the most frequent causes of genital complaints in this age group [4]. The diagnosis is clinical, based on a combination of symptoms and signs upon physical examination, including thin, pale, smooth, and shiny vaginal epithelium with diminished elasticity [2]. Estrogen supplementation (topically or systemically) is considered the most efficient treatment [5,6]. Following estrogen administration, epithelial maturation occurs, with subsequent changes in epithelial thickness, pH level, and tissue elasticity [6,7].

Treatment efficacy is evaluated in clinical trials using a range of measurements that represent the changes occurring in the vagina in response to the presence or absence of estrogen. The commonly used measures include pH level, vaginal health index (VHI, Table 1) and the vaginal maturation index (VMI) [8]. Other tools used to assess treatment are a variety of questionnaires or scores, relying on patient’s self-report, assessing symptoms’ severity or quality of life parameters. Available questionnaires include the Most Bothersome Symptoms (MBS), the Day-to-Day Impact on Vaginal Aging (DIVA) Questionnaire, the Vulvovaginal Symptoms Questionnaire (VSQ), and the Vulvar Pain Assessment Questionnaire (VPAQ) [8]. Alternatively, an array of scores rate severity of symptoms, enabling subjective comparison. Available scores include the Vaginal and Vulvar Assessment Scale, the Female Sexual Function Index (FSFI), and the Female Sexual Distress Scale-Revised (FSDS-R) [8].

As many women with GSM are reluctant to use local estrogen due to various concerns [9], alternatives for topical hormonal treatment are sought. Studies have been evaluating the effectiveness of alternative therapies that do not include estrogen, such as hydrating agents, hormonal non-estrogens including DHEA [6], and energy based treatments, such as the fractional CO_2_ laser [9,10], non-ablative vaginal Er:YAG laser (VEL) [11] and radiofrequency procedures [12].

Introduction of energy-based treatments for GSM in recent years resulted in studies evaluating the efficacy of these treatments. These modalities are claimed to improve GSM in an estrogen independent mechanism, such as connective tissue remodulation and neovascularization [9,11,12,13,14]. Nevertheless, most of the published research evaluated treatment outcomes using one or more estrogen-dependent measures (i.e., pH, VMI, and VHI). Despite reporting positive effects on symptoms and a relative improvement in the VHI index, pH levels and cytological measures, unsurprisingly, often do not show a clinically significant difference [15,16,17,18,19].

As none of the currently available parameters allow a direct and objective measurement of the vaginal condition and moisture, there is a need for such tools. Ideally, such measures should be objective, allow assessment of GSM-associated signs, and incorporate parameters relevant to new treatment modalities, which do not depend on estrogen or its effects on the vaginal tissue.

Considering patients’ reports of increased vaginal secretions following estrogen supplementation as well as following laser treatment, we opted to develop and test a vaginal dryness/moisture measurement tool. The tool we used was a calibrated filter paper strip, similar to the one used to perform the ophthalmic “Schirmer test” for eye moisture measurement.

Our goal was to evaluate the validity of this objective measurement tool for vaginal moisture by comparing measurements in symptomatic women suffering from GSM-associated vaginal dryness to measurements in healthy women without vaginal dryness.

## 2. Materials and Methods

This prospective cohort study consisted of women evaluated between January 2021 and June 2021 in an outpatient gynecologic clinic at Hadassah University Medical Center, Jerusalem, Israel. The study consisted of two groups: (1) menopausal women with complaints characteristic of GSM, including dryness, and (2) premenopausal women without vulvovaginal symptoms, who denied dryness.

An additional inclusion criterion included age > 18 years. Exclusion criteria included urinary incontinence, vaginal prolapse, vulvovaginal infection, vulvar skin disease, diagnosed Sjogren Syndrome, pregnancy, and vaginal bleeding.

The study was approved by the Institutional Review Board (Number 0923-20-HMO) and all participants signed informed consent.

Measurement of vaginal moisture was performed by placing a calibrated filter paper test strip, in a standard manner (see below) for 5 min. In all cases, to keep uniformity, the tip of the paper was located adjacent to the hymenal tissue (or its remnants) on the right side of the vaginal opening (at the “7 o’clock” location) (Figure 1).

The paper was placed using a Q-tip while the patient was lying on a gynecological bed, and it was left in place for 5 min. After 5 min, the paper strip was gently removed. Fluid amount was measured by the length of the moistened area of the strip in millimeters (Figure 2).

The test is based on the principle of capillary action, which allows the fluid from the vagina to be absorbed along the length of a paper test strip in an identical fashion as a horizontal capillary tube. The hypothesis is that the rate of travel along the test strip for 5 min time represents the amount of fluid in the vagina.

To evaluate the association between the “modified Schirmer test” and other measures, the following data were collected for each patient: vaginal pH was measured using a pH-indicator strip (pH range 3–8, Merck, Germany), VHI score was calculated (Table 1) and documented, and patients were requested to score the degree of daily vaginal dryness as well as dryness during intercourse, using a 0–10 visual analogue scale (VAS), with 0 representing no dryness, and 10 being the worst possible dryness.

Estimation of sample size was based on the expected difference in “modified Schirmer test” measurements between symptomatic menopausal women and asymptomatic non-menopausal women, calculated using available preliminary data collected for another study (NCT03063684). Given standard deviation measurements of 8.8 mm in the control group compared to 0.79 mm in the GSM group, to prove any difference of 4 mm or higher between the two groups statistically significant, with a significance of 5% (unilateral) and a power of 80%, the calculated number of women needed in each group was 30.

Data were analyzed using the SSPS software (SSPS Science, Chicago, IL, USA). Comparison of “modified Schirmer test” results between the two groups was performed using a *t*-test. Correlation analysis was performed using Pearson’s rank correlation. Statistical significance was set at *p* < 0.05.

## 3. Results

Sixty women were enrolled, and their characteristics are detailed in Table 2. Out of the 30 women who were enrolled in the control group, two were excluded from the final analysis because they did not meet the inclusion criteria, as despite verbally denying any vaginal complaints, their VAS during intercourse score was 4 or higher.

In the GSM group, 28 women were previously diagnosed with breast cancer, and 23 of them were treated with aromatase inhibitors.

“Modified Schirmer test” measurements, pH levels, VHI, and VAS are presented in Table 3. Comparison of “modified Schirmer test” measurements between the two study groups is presented in Table 3 and Figure 3. The differences between the measurements in the study group and the control group were statistically significant.

A statistically significant difference was observed between the two groups regarding the number of vaginal deliveries. We therefore performed a covariate analysis, correcting for vaginal deliveries, to establish that the distinction in the “modified Schirmer test” measurements was not a result of the difference in vaginal deliveries alone. The analysis yielded a statistically significant difference after the correction as well.

Correlations between all measured parameters of the entire study population, expressed by Pearson’s correlation coefficient, are shown in Table 4. The modified Schirmer measurements showed strong correlations to pH, VHI, and intercourse-VAS. All correlations were statistically significant (*p* < 0.001).

## 4. Discussion

The purpose of this study was to test a new, objective measurement modality for vaginal moisture/dryness and to evaluate its validity. To achieve this, we used a calibrated filter-paper test strip, similar to the one used for the ophthalmic Schirmer test, in a standardized manner comparing symptomatic women suffering from GSM-associated vaginal dryness to women without dryness. The comparison yielded a statistically significant difference between the groups, suggesting that this test is correlated with symptoms and is useful in distinguishing symptomatic vaginal dryness from normal vaginal moisture.

Furthermore, the “modified Schirmer test” measurements showed strong and statistically significant correlations to the currently used indices of vaginal atrophy, i.e., pH, VHI, and VAS scores. These correlations imply a non-inferiority of the test compared to currently accepted measures.

It is important to note that the two study groups were significantly diverse in most characteristics (age, hormonal status, contraception, and lubricant use), as they represent essentially different phases of women’s lives in terms of hormonal status.

We found no statistically significant distinction in the “modified Schirmer test” measurements between women who reported using hormonal contraceptives and those who reported using non-hormonal contraceptives; neither did we find differences between women with diverse hormonal statuses in the control group (i.e., normal menstruation, amenorrhea associated with hormonal IUD, and perimenopause). This could be a resultant of the small sample size, as these were subgroups of the entire control group. Although larger studies are needed to confirm our findings, the finding that this test is not altered by hormonal status or contraceptive modalities may indicate yet another advantage of its use, as it provides a direct and hormonal-independent indication of the vaginal moisture level.

Clinical trials evaluate treatment efficacy for GSM mostly using measures that represent vaginal changes occurring in response to the presence or absence of estrogen. Most frequently, these include pH level, the VHI, and the VMI [13]. Vaginal pH level is mainly determined by the presence or absence of lactobacilli. These bacteria produce lactic acid as they catabolize glycogen, which in turn decreases the vaginal pH level to 3.5–4.5. As glycogen is found in mature epithelial cells yet absent in para-basal epithelial cells (which represent atrophy), a vaginal pH level >5 in the absence of other causes (such as infection or presence of semen, cervical mucous, or blood) is indicative of glycogen absence, thus, of decreased estrogen and resultant atrophy [3,8]. The VHI, first described by Bachmann in 1992 [20], is widely used even though it has not been validated (Gloria Bachmann, personal communication) and uses mostly subjective parameters. The VHI is comprised of five parameters, observed during examination per speculum, graded from 1 to 5 each. Four parameters are not well defined and are subjective to inter and intra observer differences: vaginal elasticity, fluid volume (measured by fluid pooling in the fornix), epithelial integrity, and epithelial moisture, whereas only one measurement is objective—the vaginal pH [21]. According to the VHI, atrophy is defined as a score lower than 15 [21]. The VHI does not include clear examination instructions and is, therefore, dependent on examiner’s interpretation, thus lacking uniformity. Like other measures used, it is, at least partially, an estrogen-dependent index, as pH level comprises one fifth of its value.

VMI is another tool that represents vaginal estrogen influence on epithelial cytology, through calculating the relative percentage of superficial (mature) cells compared to intermediate and para-basal epithelial cells [3,8].

The main strength of our study is the introduction of a new measurement, which is easy to use and interpret. To our knowledge, previous studies that aimed to test the amount of vaginal discharge included usage of swabbing the entire vagina during an exam and weighing the swab [22], weighing tampons after inserting them for 8 h [23], by aspiration of vaginal fluid [24], or by pad weighing [25]. Nevertheless, these modalities are either effort- and time-consuming or lack precision, as the weighed fluid is not necessarily comprised of vaginal discharge alone and may also contain sweat or urine. Other drawbacks include patient discomfort and functionality. In addition, none of these measures were studied in relation to vaginal atrophy.

Furthermore, the current tool may allow an objective and direct evaluation of dryness symptoms when there is a discrepancy between symptoms and findings. For example, in patients who complain of dryness but have an apparently normal examination, this tool may allow confirmation of a normal amount of discharge, suggesting a sensory issue or vulvodynia and directing further evaluation or suitable treatment.

Our study is limited for several reasons, including the small number of participants, the lack of measures among asymptomatic menopausal women, and the lack of measurement comparison before and after treatment. The “modified Schirmer test” should be further evaluated in larger scale studies, which will generate more accurate and specific ranges of measurements, representing and distinguishing between menopause-associated-vaginal-atrophy, menopause without atrophy, and an estrogenized state. Furthermore, larger sample sizes may allow the division of measurement ranges into subranges, such as pre-menopausal women using hormonal contraceptives with and without vaginal dryness, compared to age-matched women who are not using hormonal contraceptives.

The main advantage of the “modified Schirmer test” is its potential to serve as an objective test in assessing vaginal dryness/moisture level in relation to non-estrogenic treatments for vaginal dryness. Therefore, the validity of this test should be further studied by comparison of measurements before and after treatments aimed to relieve vaginal atrophy.

## 5. Conclusions

Our findings suggest that the “modified Schirmer test” can be used as an objective indicator of vaginal moisture level, distinguishing women who suffer from vaginal dryness from those who do not. This test may also prove useful for evaluation of non-hormonal treatment results in longitudinal research, where direct and objective measures of vaginal moisture are sought to complement the subjective VAS score.

## Figures and Tables

**Figure 1 diagnostics-12-00574-f001:**
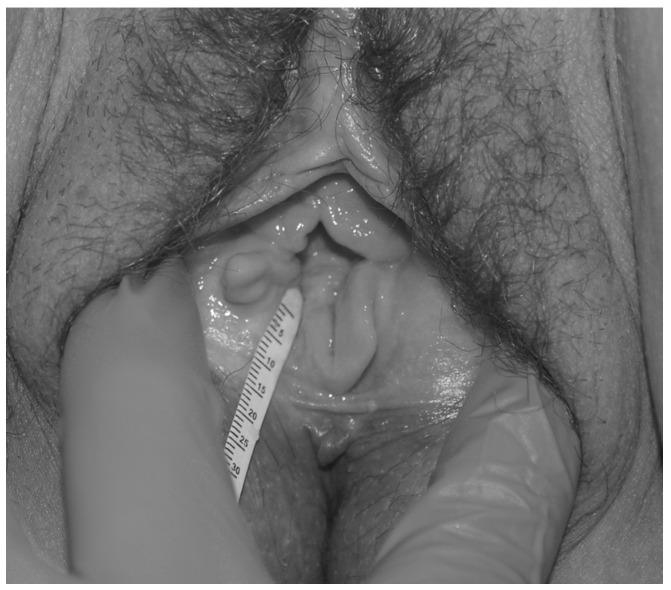
Modified Schirmer test—paper strip location. Measurement of vaginal moisture is performed by placing the tip of a calibrated filter paper test strip adjacent to the remnants of the hymenal tissue on the right side of the vaginal opening, at the “7 o’clock” location.

**Figure 2 diagnostics-12-00574-f002:**
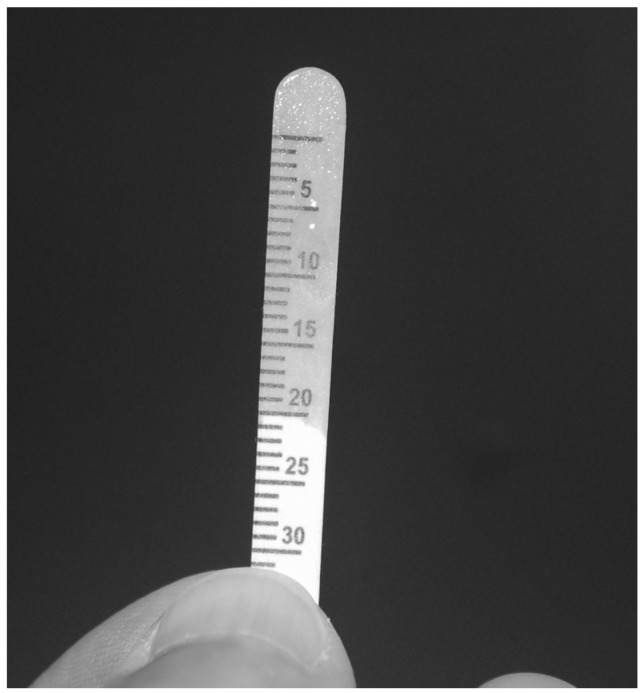
Modified Schirmer test paper strip measurement modality. Fluid amount is measured by the length of the moistened area of the strip in millimeters, i.e., 20 mm.

**Figure 3 diagnostics-12-00574-f003:**
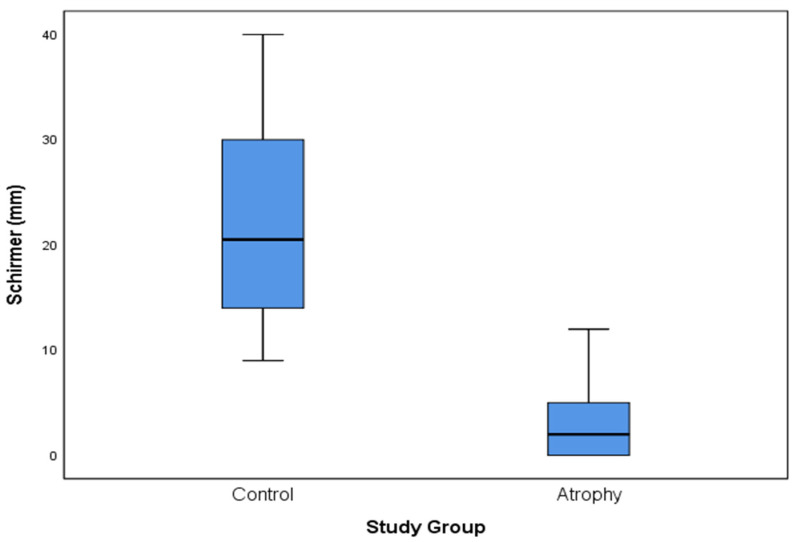
Modified Schirmer measurements in control and study groups. The figure graphically presents the differences between “modified Schirmer test” measurements in the control and the GSM groups. The horizontal line within the box indicates the median, boundaries of the box indicate the 1st and 3rd quartile, and whiskers indicate the minimum and maximum values.

**Table 1 diagnostics-12-00574-t001:** The vaginal health index (VHI) (adapted from Bachmann et al. [20]).

Score	1	2	3	4	5
Elasticity	None	Poor	Fair	Good	Excellent
Fluid Volume (Pooling of Secretions)	None	Scant amount, vault not entirely covered	Superficial amount, vault entirely covered	Moderate amount of dryness (small areas of dryness on cotton-tip applicator)	Normal amount (fully saturates on cotton-tip applicator)
pH	6.1 or above	5.6–6.0	5.1–5.5	4.7–5.0	4.6 or below
Epithelial Integrity	Petechiae noted before contact	Bleeds with light contact	Bleeds with scraping	Not friable, thin epithelium	Normal
Moisture (Coating)	None, surface inflamed	None, surface not inflamed	Minimal	Moderate	Normal

**Table 2 diagnostics-12-00574-t002:** Patients’ characteristics.

Study Group		Control (N = 28)	GSM (N = 30)
Age	Mean (Std. Deviation)	38.2 (6.7)	48.3 (7.1)
	Median (Range)	36.5 (28–52)	47.5 (36–63)
Gravidity	Mean (Std. Deviation)	3.1 (1.9)	4.1 (2.1)
	Median (Range)	3 (0–9)	4 (2–10)
Vaginal Deliveries	Mean (Std. Deviation)	1.8 (1.4)	3 (1.3)
	Median (Range)	2 (0–4)	3 (1–8)
Hormonal Status	Menstruation (%)	19 (67.9)	0 (0)
	Amenorrhea with Hormonal IUD (%)	7 (25)	0 (0)
	Perimenopause (%)	2 (7.1)	0 (0)
	Menopause (%)	0 (0)	30 (100)
Contraception	None (%)	8 (28.6)	30 (100)
	BTL (%)	2 (7.1)	0 (0)
	Condoms (%)	4 (14.3)	0 (0)
	Copper IUD (%)	4 (14.3)	0 (0)
	Hormonal IUD (%)	7 (25)	0 (0)
	HCs (%)	3 (10.7)	0 (0)

GSM = genitourinary syndrome of menopause, IUD = intra-uterine device, BTL = bilateral tubal ligation, HCs = hormonal contraceptives.

**Table 3 diagnostics-12-00574-t003:** Measurements of parameters evaluating vaginal atrophy.

Study Group	Control (N = 28)		Atrophy (N = 30)		*t*-Test for Equality of Means
	Mean (Std. Deviation)	Median (Range)	Mean (Std. Deviation)	Median (Range)	
Modified Schirmer (mm)	21.7 (9.3)	20.5 (9–40)	3.3 (3.9)	2 (0–12)	*p* < 0.001
pH	4.1 (0.4)	4 (4–6)	6.9 (0.85)	7 (5–8)	*p* < 0.001
VHI	24.3 (1.0)	25 (22–25)	12.3 (2.3)	12.5 (8–18)	*p* < 0.001
VAS daily dryness	0.4 (0.8)	0 (0–3)	6.0 (3.8)	7.5 (0–10)	*p* < 0.001
VAS intercourse dryness	0.9 (1.1)	0 (0–3)	9.7 (0.5)	10 (8–10)	*p* < 0.001

**Table 4 diagnostics-12-00574-t004:** Correlations between the different measures. Pearson’s correlation coefficient is a measure of the linear correlation between these parameters. Correlation coefficients of 1 or −1 represent perfect correlations, whereas a correlation coefficient of 0 represents no correlation. Thus, the closer to 1 or −1 the value of the coefficient is, the stronger the correlation.

		Modified Schirmer (mm)	pH	VHI	VAS Daily Dryness
**Correlation Coefficient (Significance**)	pH	−0.714 (0.0)			
	VHI	0.775 (0.0)	−0.901 (0.0)		
	VAS daily dryness	−0.544 (0.0)	0.711 (0.0)	−0.709 (0.0)	
	VAS intercourse dryness	−0.821 (0.0)	0.885 (0.0)	−0.936 (0.0)	0.731 (0.0)

## Data Availability

The data presented in this study are available on request from the corresponding author.

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
