# Peer review of "“Modified Schirmer Test” as an Objective Measurement for Vaginal Dryness: A Prospective Cohort Study"

_diagnostics, 2022, doi:10.3390/diagnostics12030574_

Round 1
Reviewer 1 Report
The authors demonstrate that a calibrated filter paper strip could be used for the quantification of vaginal fluid which was used in a Schirmer test ie. used for eye moisture. They aimed to evaluate this test in in symptomatic women with Genitourinary Syndrome of Menopause to those of women without vaginal dryness. They show significant different between the groups They also correlations between the modified Schirmer test and VAS score. They argue this test maybe useful for evaluation of non-hormonal treatments aimed to treat vaginal dryness.
This is a interesting paper.
Author Response
We would like to thank the reviewer for the review and the kind remarks.
Reviewer 2 Report
Dear authors, I congratulate with your work. The topic is of main interest, dealing with a minimally invasive test to objectively evidence the vaginal moisture status. I have two main issues that, in my opinion, may limit the validity of this study.
The first is a practical clinical issue. The proposed test (that requires 5 minutes) statistically correlates with the VAS score (that requires few seconds): how do you think this test may be useful in the clinical practice?
The second issue is methodological. You included in the study population patients in menopausal status and with GSM symptoms, and in the control group patients in menstrual or pre-menopausal status and without GSM symptoms. How can you be sure that the test correlates only with GSM symptoms and not also with menopausal status. Comparing symptomatic and asymptomatic menopausal patients would be much more useful, in my opinion. You stated this in the limitations of your study, but I think that this issue may limit the results of the work: study population and control group differ in a significant aspect.
Author Response
We would, first, like to thank the reviewer for the insightful remarks.
Regarding the issues addressed:
The first is a practical clinical issue. The proposed test (that requires 5 minutes) statistically correlates with the VAS score (that requires few seconds): how do you think this test may be useful in the clinical practice?
The VAS is a subjective measure, which relies on self-report and patient's perception of her current state. The need for an objective measurement-tool such as the one we have presented, raised while we were conducting another study, in which we analyzed the efficiency of energy-based treatment for GSM in breast cancer survivors (NCT03063684). In that study, we noticed that the main improvement reported by patients following intervention was the amount of vaginal discharge, which was not equally represented by any of the available and commonly used tools, such as changes in pH or VMI. We have therefore developed this measurement device mainly as a research parameter that can facilitate the evaluation of treatment efficacy, comparing results before and after treatment, by quantifying vaginal fluid amount. Our interim results show a significant difference between measurements before and after the treatment (i.e., in the same patient), supporting the efficacy of the energy-based treatment in improving vaginal dryness but not for other GSM associated symptoms. This test, used as an objective tool, enhances the validity of the VAS reports, as it can demonstrate that the benefit is not only apparent in patients' reported subjective symptoms but can also be measured objectively. Differentiating treatment's effects using objective measures can also assist in better understanding of the mechanism of action – improvement in fluid amount alone can imply that neovascularization is causing fluid extravasation, thus improving symptoms of dryness in an estrogen-independent mechanism.
In the clinical practice it may be useful in several ways; When treating GSM, our aim as physicians is to improve patients' symptoms. Thus, indeed, the VAS score which represents symptomatic improvement is the best indicator. Nevertheless, the current tool may allow an objective and direct evaluation of dryness-symptoms when there is a discrepancy between symptoms and findings. For example, in patients who complain of dryness albeit an apparently normal examination, this tool may allow confirmation of a normal amount of discharge, suggesting a sensory issue or vulvodynia and directing further evaluation or suitable treatment (added in the discussion, page 7, lines 216-220). In addition, it may allow an objective evaluation of patients' status following treatment, allowing differentiation of responsive versus nonresponsive patients, estimating need for further treatment, while avoiding empiricism.
The second issue is methodological. You included in the study population patients in menopausal status and with GSM symptoms, and in the control group patients in menstrual or pre-menopausal status and without GSM symptoms. How can you be sure that the test correlates only with GSM symptoms and not also with menopausal status. Comparing symptomatic and asymptomatic menopausal patients would be much more useful, in my opinion. You stated this in the limitations of your study, but I think that this issue may limit the results of the work: study population and control group differ in a significant aspect.
We completely agree with the reviewer's remark. This is a pilot study, in which we have compared essentially different populations, as the reviewer rightfully noted, in order to test for validity. We aimed to achieve an initial validation of this measurement device, for which we have chosen two significantly different study groups.
After finding that this tool can differentiate symptomatic dryness from normal vaginal moisture, we are now planning a follow-up study which will compare symptomatic and asymptomatic menopausal women, just as the reviewer suggested. As menopausal women are a heterogenous group, approximately half of which do not report symptoms of GSM, correlating complaints of dryness to objective measurements is an interesting topic to investigate, possibly allowing better understanding of this heterogenicity (i.e., whether absence of symptoms results from absence of dryness, or absent albeit an objective finding of dryness).
Round 2
Reviewer 2 Report
Dear Authors,
Thank you for having taken into account my suggestions. I agree with your reply.